# Plasma Based Protein Signatures Associated with Small Cell Lung Cancer

**DOI:** 10.3390/cancers13163972

**Published:** 2021-08-06

**Authors:** Johannes F. Fahrmann, Hiroyuki Katayama, Ehsan Irajizad, Ashish Chakraborty, Taketo Kato, Xiangying Mao, Soyoung Park, Eunice Murage, Leona Rusling, Chuan-Yih Yu, Yinging Cai, Fu Chung Hsiao, Jennifer B. Dennison, Hai Tran, Edwin Ostrin, David O. Wilson, Jian-Min Yuan, Jody Vykoukal, Samir Hanash

**Affiliations:** 1Department of Clinical Cancer Prevention, The University of Texas M. D. Anderson Cancer Center, Houston, TX 77030, USA; jffahrmann@mdanderson.org (J.F.F.); HKatayama1@mdanderson.org (H.K.); EIrajizad@mdanderson.org (E.I.); ashishchak@utexas.edu (A.C.); TKato1@mdanderson.org (T.K.); XMao2@mdanderson.org (X.M.); SNPark@mdanderson.org (S.P.); ENMurage@mdanderson.org (E.M.); LAMartin1@mdanderson.org (L.R.); CYu3@mdanderson.org (C.-Y.Y.); YCai4@mdanderson.org (Y.C.); FCHsiao@mdanderson.org (F.C.H.); jbdennis@mdanderson.org (J.B.D.); jvykouka@mdanderson.org (J.V.); 2Department of Thoracic-Head & Neck Medical Oncology, The University of Texas M. D. Anderson Cancer Center, Houston, TX 77030, USA; htran@mdanderson.org; 3Department of Pulmonary Medicine, The University of Texas M. D. Anderson Cancer Center, Houston, TX 77030, USA; EJOstrin@mdanderson.org; 4Division of Pulmonary, Allergy and Critical Care Medicine, School of Medicine, University of Pittsburgh, Pittsburgh, PA 15213, USA; wilsondo@upmc.edu; 5Division of Cancer Control and Population Sciences, UPMC Hillman Cancer Center, University of Pittsburgh, Pittsburgh, PA 15232, USA; yuanj@upmc.edu; 6Department of Epidemiology, Graduate School of Public Health, University of Pittsburgh, Pittsburgh, PA 15261, USA

**Keywords:** small-cell lung cancer, proteomics, biomarkers

## Abstract

**Simple Summary:**

Small-cell lung cancer (SCLC) typically presents at an advanced stage and is associated with high mortality. When diagnosed at an early stage with localized disease, long-term survival can, however, be achieved. In this study, we report a comprehensive proteomic profiling of case plasmas collected at the time of diagnosis or preceding diagnosis of SCLC with the objective of identifying blood-based markers associated with disease pathogenesis. Our study reveals the occurrence of circulating protein features centered on signatures of oncogenic MYC and YAP1 that were elevated in plasmas of cases at and before the time-of-diagnosis of SCLC. We further report several proteins, particularly inflammatory markers, that were identified as elevated in plasma several years prior to the diagnosis of SCLC and that may indicate increased risk of disease. In summary, our study identifies several novel circulating proteins associated with SCLC development that may offer utility for early detection.

**Abstract:**

Small-cell-lung cancer (SCLC) is associated with overexpression of oncogenes including Myc family genes and YAP1 and inactivation of tumor suppressor genes. We performed in-depth proteomic profiling of plasmas collected from 15 individuals with newly diagnosed early stage SCLC and from 15 individuals before the diagnosis of SCLC and compared findings with plasma proteomic profiles of 30 matched controls to determine the occurrence of signatures that reflect disease pathogenesis. A total of 272 proteins were elevated (area under the receiver operating characteristic curve (AUC) ≥ 0.60) among newly diagnosed cases compared to matched controls of which 31 proteins were also elevated (AUC ≥ 0.60) in case plasmas collected within one year prior to diagnosis. Ingenuity Pathway analyses of SCLC-associated proteins revealed enrichment of signatures of oncogenic MYC and YAP1. Intersection of proteins elevated in case plasmas with proteomic profiles of conditioned medium from 17 SCLC cell lines yielded 52 overlapping proteins characterized by YAP1-associated signatures of cytoskeletal re-arrangement and epithelial-to-mesenchymal transition. Among samples collected more than one year prior to diagnosis there was a predominance of inflammatory markers. Our integrated analyses identified novel circulating protein features in early stage SCLC associated with oncogenic drivers.

## 1. Introduction

Small-cell lung cancer (SCLC) is a highly lethal malignancy that generally presents at an advanced stage and is associated with neuroendocrine phenotypic features [1,2]. When diagnosed at an early stage with localized disease, long-term survival can be achieved [3].

The National Lung Cancer Screening Trial (NLST) findings indicate screening with LDCT can reduce mortality due to lung cancer by 20%, with similar results since reported from the NELSON trial [4,5].Yet, CT-based screening in NLST was not found effective for detecting small-cell lung cancer (SCLC) at an early stage, with SCLC often being detected as an interval cancer [6], without a survival improvement amongst these subjects [7]. Thus, there remains a clinical need to develop biomarkers to enable detection of SCLC at an early stage to improve the potential for longer survival.

Increasing evidence highlights utility of liquid-biopsies as an ideal ‘minimally invasive’ approach for early interception of disease by identifying of those individuals who are at high risk of either developing or harboring disease and thereby triggering clinical follow-up such as LDCT [8,9,10,11]. In prior studies of lung adenocarcinoma, we identified a circulating protein signature that reflected activation at early stages of Titf1/Nkx2-1, a known lineage-survival oncogene in lung cancer. The signature notably included the immature form of surfactant protein B [12]. Subsequent validation studies provided evidence that a biomarker panel including prosurfactant protein B may improve lung cancer risk assessment [8]. However, there remains a need to uncover biomarker signatures that reflect subtypes of lung cancer at early stages. To date, several blood-based markers have been identified in association with SCLC, including neuron specific enolase (NSE), Progastrin-releasing peptide (ProGRP), chromogranin A (CgA) and pro-opiomelanocortin (POMC) [13,14,15,16,17,18,19]. Performance of these markers in the early stage setting is limited because of reduced expression [14,16,17].

Molecular characterization of tumor tissues has identified key determinants associated with SCLC development and progression including loss of TP53 and RB1, MYC copy number amplification, and activation of the PI3K/AKT/mTOR pathway [20,21,22,23]. More recent evidence has defined a new classification model of SCLC subtypes, characterized by differential expression of the transcriptional regulators achaete-scute homologue 1 (ASCL1), neurogenic differentiation factor 1 (NeuroD1), yes-associated protein 1 (YAP1) and POU class 2 homeobox 3 (POU2F3) [24].

In this study, we performed comprehensive proteome profiling of plasmas collected before the diagnosis of SCLC and plasmas from subjects with newly diagnosed early stage SCLC and compared findings to proteomic profiles from matched healthy controls to assess the potential association of protein changes in circulation with oncogenic drivers in SCLC. Findings were further integrated with proteomic profiles of conditioned media from 17 SCLC cell lines and gene expression data from SCLC tumors.

## 2. Materials and Methods

### 2.1. Human Specimen

All human blood samples were obtained following institutional review board approval, and patients provided written informed consent.

The initial discovery set consisted of EDTA-plasma samples from 15 newly diagnosed early stage SCLC cases and 15 controls matched on sex, age and smoking history from the University of Texas MD Anderson Cancer Center (MDACC) (Table 1). Case plasmas were obtained from participants in the Genomic Marker-Guided Therapy Initiative (GEMINI) project (IRB protocol PA13-0589). The GEMINI project entails detailed clinical and molecular information of over 4000 lung cancer patients as well as a biorepository for plasma samples. Control plasmas were selected from participants in the Lung cancer Early Detection Assessment of risk, and Prevention (LEAP) study (IRB protocol 2013-0609). The LEAP cohort includes 586 participants enrolled at MD Anderson who were eligible for low-dose CT screening based on United States Preventative Services Task Force (USPSTF) 2013 criteria. Control plasmas were selected from participants that were confirmed to be cancer-free for a minimum of four years following blood draw.

The pre-clinical cohort consisted of pre-diagnostic plasmas from 15 SCLC subjects diagnosed within a median of 2.4 years of blood draw along with 15 controls with no history of cancer during the period of follow-up. Samples were derived from participants in the Pittsburgh Lung Screening Study (PLuSS) [25] and Singapore Chinese Health Study (SCHS) [26]. Controls were matched based on age, sex and smoking status (Table 2).

The PLuSS cohort recruiting criteria were: (1) age 50 to 79 years; (2) no personal lung cancer history; (3) nonparticipation in concurrent lung cancer screening studies; (4) no chest computed tomography (CT) within 12 months; (5) current or ex-cigarette smoker of at least one-half pack per day for at least 25 years, and, if they quit smoking, quit for no more than 10 years before study enrollment; and (6) body weight less than 400 pounds from January 2002 [25].

The SCHS enrolled a total of 63,257 Chinese persons aged 45–74 years between 1993 and 1998. Participants belonged to one of the major dialect groups (Hokkien or Cantonese) of Chinese in Singapore and were citizens or permanent residents of government-built housing estates, where 86% of the general population resided during the enrollment period. Cancer diagnoses and deaths in this cohort were identified via linkage with the Singapore Cancer Registry and the Singapore Registry of Births and Deaths [26].

### 2.2. SCLC Cell Line-Derived Conditioned Media

Seventeen SCLC cell lines (H1607, HCC4002, H209, H211, H2195, H2679, H345, H524, H526, H69P, H69AD, H82, HCC4001, HCC4003, HCC4004, HCC4005, H1048) representative of the consensus SCLC subtypes were analyzed (Appendix B Table A1) [24].

Collection of conditioned media for protein analysis was performed as previously described [12]. Briefly, SCLC cell lines were grown in RPMI1640 (Pierce) containing 10% of dialyzed fetal bovine serum (FBS) (Invitrogen), 1% penicillin/streptomycin cocktail and 13C-lysine (Cambridge Isotope Laboratories, #CLM-2247-H) for 7 passages in accordance with the standard SILAC protocol [27]. The purpose of 13C-lysine labeling was to enable discrimination between SCLC released proteins and proteins that occur in FBS. Whole cell extracts of cells were prepared by sonication of ~2 × 10^7^ cells in 1 mL of Tri-HCl buffer (pH 8.0) containing detergent octyl-glucoside (OG) (1% *w/w*), 4M urea, 3% isopropanol and protease inhibitors (complete protease inhibitor cocktail, Roche Diagnostics, Germany) followed by centrifugation at 20,000× *g*. Secreted and shed proteins were obtained directly from the cell conditioned media with 0.1% dialyzed FBS after 48 h of culture. Cells and debris were removed by centrifugation at 5000× *g* and filtration through a 0.22 μm filter.

### 2.3. Mass Spectrometry Analyses of Human Plasmas

Plasma volumes of 100 μL were processed using immuno-depletion affinity column Hu-14 10 × 100 mm (Agilent Technologies, Santa Clara, CA USA, #5188-6559) to remove 14 high abundance plasma proteins: Albumin, IgG, IgA, Transferrin, Haptoglobin, Fibrinogen, α1-Antitrypsin, α1-Acid Glycoprotein, Apolipoprotein AI, Apolipoprotein AII, Complement C3, Transthyretin, IgM and α2-Marcroglobulin. The flow-through fraction was then used for profiling the lower abundance free (non-Ig bound) plasma proteome. To prepare for proteomics analysis, samples were concentrated and reduced with TCEP and alkylated by 2-chloro-N,N-diehtylcarbamidomethyl (diethylcarbamidomethyl). Next, the buffer was exchanged to TEAB and trypsin digested, 100 μg corresponding peptides from each pool was desalted by C18-CX Monospin column (GL Sciences, Torrance, CA, USA) and dried by SpeedVac (Thermo Scientific, Waltham, MA, USA). Each of the dried pool was individually dissolved and labeled with 10 plex Lys-TMT Channel (Thermo Scientific, #90309) and combined, fractionated into 12 fractions with alkaline 0.1% Triethylamine/acetonitrile reversed phase mode using C18 Monospin Large column (GL Sciences, Torrance, CA, USA). The step elution was done by B concentration of 20%, 25%, 30%, 35%, 40%, 45%, 50%, 55%, 60%, 70%, 80% and 100% using mobile phase A (0.1% Triethylamine in Water/acetonitrile 98/2) and Mobile phase B (0.1% Triethylamine in water/acetonitrile 5/95), then the fractions were dried by the SpeedVac (Thermo Scientific).

The samples were subsequently reconstituted with acetonitrile/water/trifluoroacetic acid (TFA) (2:98:0.1, *v/v/v*) and individually analyzed by Easy nanoLC 1000 system (Thermo Scientific, Waltham, MA, USA) coupled Q-exactive mass spectrometer using a 15 cm column (75 μm ID, C18 3 μm, column Technology Inc) as a separation column, and Symmetry C18 180 um ID × 20 mm trap column (Waters Inc., Milford, MA, USA) over a 120 min gradient. Mass spectrometer parameters were spray voltage 3.0 kV, capillary temperature 275 °C, Full scan MS of scan range 350–1800 m/z, Resolution 70,000, AGC target 3e6, Maximum It 50 msec and Data dependent MS2 scan of resolution 17,500 in profile mode, AGC target 1e5, Maximum IT 100 msec and repeat count 10 in HCD mode.

Acquired mass spectrometry data were processed by Proteome Discover 1.4 (Thermo Scientific). The tandem mass spectra were searched against Uniprot human database 2017 using Sequest HT. The modification parameters were as follows: fixed modification of Cys alkylated with diethylcarbamidomethyl (+113.084064), Lys with 10 plex TMT (+229.162932, N-terminal and Lys), and variable modification of Methionine oxidation (+15.99491). The precursor mass tolerance of the parent and fragment mass were 10 ppm and 0.02 Da, respectively. Searched data was further processed with the Target Decoy PSM Validator function with a false-discovery rate (FDR) of 0.05.

### 2.4. Mass Spectrometry Analyses of SCLC Cell Line Conditioned Media

SCLC cell line conditioned media were concentrated and reduced with TCEP and alkylated by acrylamide (propionamide) and the intact proteins were fractionated into 14 fractions by AQUITY UPLC system (Waters Inc., Milford, MA, USA) in reversed phase mode using a RPGS reversed-phase column (4.6 mm × 150 mm, 15 μm particle, 1000 Å, Column Technology Inc, Fremont, CA, USA) and dried, trypsin digested and subjected to mass spectrometry analysis. The tryptic peptides were analyzed by NanoAcquity UPLC system coupled to WATERS SYNAPT G2-Si mass spectrometer using 15 cm column (75 μm ID, C18 3um, Column Technology Inc, Fremont, CA, USA) as a separation column, and Symmetry C18 180 μm ID × 20 mm trap column (Waters Inc., Milford, MA, USA) over a 120 min gradient. LC HDMSE data were acquired in resolution mode with SYNAPT G2-Si using Waters Masslynx (version 4.1, SCN 851, Waters Inc). The capillary voltage was set to 2.80 kV, sampling cone voltage to 30 V, source offset to 30 V and source temperature to 100 °C. Mobility utilized high-purity N2 as the drift gas in the IMS TriWave cell. Pressures in the helium cell, Trap cell, IMS TriWave cell and Transfer cell were 4.50 mbar, 2.47 × 10^−2^ mbar, 2.90 mbar and 2.53 × 10^−3^ mbar, respectively. IMS wave velocity was 600 m/s, helium cell DC 50 V, Trap DC bias 45 V, IMS TriWave DC bias V and IMS wave delay 1000 μs. The mass spectrometer was operated in V-mode with a typical resolving power of at least 20,000. All analyses were performed using positive mode ESI using a NanoLockSpray source. The lock mass channel was sampled every 60 s. The mass spectrometer was calibrated with a (Glu1) fibrinopeptide solution (300 fmol/μL) delivered through the reference sprayer of the NanoLockSpray source. Accurate mass LC-HDMSE data were collected in an alternating, low energy (MS) and high energy (MSE) mode of acquisition with mass scan range from m/z 50 to 1800. The spectral acquisition time in each mode was 1.0 s with a 0.1-s inter-scan delay. In low energy HDMS mode, data were collected at a constant collision energy of 2 eV in both Trap cell and Transfer cell. In high energy HDMSE mode, the collision energy was ramped from 25 to 55 eV in the Transfer cell only. The RF applied to the quadrupole mass analyzer was adjusted such that ions from m/z 300 to 2000 were efficiently transmitted, ensuring that any ions observed in the LC-HDMSE data less than m/z arose from dissociations in the Transfer collision cell. The acquired LC-HDMSE data was subsequently processed and searched against the Uniprot proteome database (Human, January 2017) through the ProteinLynx Global Server (PLGS, Waters Inc., Milford, MA, USA) with two trypsin miss cleavage allowed. The modification search settings included cysteine (Cys) alkylation with propionamide (+71.03714) as a fixed modification, and methionine (Met) oxidation (+15.99491) as a variable modification. The searched data was filtered with a False Discovery Rate 4%.

The total number of spectral counts for each protein group output was used for semi-quantitative analyses. Each dataset was normalized to the total number of spectral counts. To mitigate potential bias of proteins manifest in conditioned media as a result of cell turnover, we employed the spectral counting method to estimate protein enrichment in conditioned media as previously described [12]. Briefly, enrichment in conditioned media was calculated as follows: [(C_x p_/Nf) + 1]/(C_te p_ + 1). C_x_ represents spectral counts of each protein (p) in the conditioned media; Nf is the normalization factor (total spectral counts in the conditioned media/total spectral counts in the whole extract); Cte is the spectral counts for the same protein (p) observed in the whole cell extract. Here, we considered a protein to be enriched in SCLC cell line conditioned medium if the enrichment ratio ≥2.0.

### 2.5. Ingenuity Pathway Enrichment Analysis

Ingenuity Pathway Enrichment Analysis (IPA) [28] was conducted on proteins identified by mass spectrometry that were elevated in plasmas of patients that later developed SCLC or in plasmas of patients with newly diagnosed early stage SCLC as well as proteins that were elevated in case plasmas and that were also quantified in SCLC cell line conditioned media. Statistical significance of enriched pathways was determined by 2-sided Fisher’s Exact Test. Subcellular localization as well as functional subtyping of protein features were derived from IPA; proteins annotated as being secreted was downloaded from The Human Protein Atlas (https://www.proteinatlas.org/humanproteome/cell/secreted+proteins, accessed on the 3 March 2021). ChIP Enrichment Analysis (ChEA) transcriptional targets of c-MYC and YAP1 were downloaded from Harmonizome database [29,30]. For MYC, we utilized the MYC-20876797 Human Medulloblastoma gene set.

### 2.6. Statistical Analysis

For human plasmas, model discrimination was assessed by area under curve (AUC) of the receiver operating characteristic curve (ROC). ROC analyses were performed using pROC (version 1.15.3) in the R software environment (version 3.6.1, The R Foundation). The 95% confidence intervals (CI) for AUCs were estimated using Delong method [31]. *p*-values are reported based on 2-sided Wilcoxon rank sum test unless otherwise specified. We chose to focus on AUC to evaluate the distribution of data points and to assess the ability of features to distinguish cases from controls. Herein, we selected those markers that were elevated in cases using a cutoff of AUC greater than 0.60. Odds ratio was used to assess the performance for pre-diagnosis cases where their respective diagnosis time are more than 1 year. Markers with odds ratio greater and equal to 2 was selected as important markers.

## 3. Results

### 3.1. Proteomic Profiling Reveals Signatures Associated with Oncogenic Drivers Manifest in Plasmas at Early Stages of SCLC

To identify plasma-based signatures for early stage SCLC, we performed comprehensive proteomic profiling of individual plasmas collected from 15 newly diagnosed early stage SCLC cases (6 stage I and 9 stage II) and 15 controls matched based on age, sex and smoking pack years (Table 1). Using a threshold of identification in at least 50% of the samples, a total of 1587 plasma proteins were quantified. Differential analyses yielded 272 proteins with AUC estimates of ≥0.60 for differentiating SCLC cases from matched controls, which included the SCLC-associated marker ENO2/NSE (Figure 1A and Appendix A) [14,16,19]

Ingenuity pathway analysis (IPA) of the 272 proteins revealed YAP1 and related transcriptional enhanced associate domain transcription factor (TEAD)-family members [32] among top predicted transcriptional regulators, in addition to MYC as a central network node (Figure 1B,C). To this end, intersection of protein findings with ChEA transcriptional targets of MYC and YAP1 revealed 15 of the 272 proteins to be known transcriptional targets of c-MYC whereas another 49 (18.0%) are known transcriptional targets of YAP1 (Appendix A).

Among the 15 MYC-associated proteins were several proteins involved in epithelial-to-mesenchymal transition (EMT), including CIT, DIAPH3, FBXO11, THBS1 and YWHAZ (Appendix B Table A2). YAP1-associated proteins were biologically linked to actin cytoskeletal signaling and epithelial adherens junction signaling (Appendix B Table A3) [33,34]. Moreover, several cytoskeletal associated-proteins including ACTB, ATCBL2, ACTC1, COTL1, FERMT3, PFN1, TPM3, TPM4 and VCL were found to be elevated in case plasmas compared to matched control (Appendix A), that may reflect cytoskeletal reorganization and degradation of the extracellular matrix (ECM) that occurs during EMT [35,36,37].

### 3.2. Intersection of SCLC-Associated Protein Signatures between Human and SCLC Cell Line-Derived Conditioned Medium

We searched proteins released into conditioned media (CM) of 17 SCLC cell lines to determine the overlap with proteins that were elevated (AUC ≥ 0.60) in plasmas of early stage SCLC cases and to assess whether SCLC tumor cells potentially contribute to increased levels of some of these proteins. A total of 641 proteins were quantified in SCLC cell line CM based on a threshold of identification in at least 50% of the cell lines and an enrichment ratio >2.0 (see Section 2) (Appendix A). Of these, 28 overlapped with proteins that were also found to be elevated (AUC ≥ 0.60) in case plasmas (Figure 1D). Of note, among the overlapping proteins were several of the YAP1 transcriptional targets (ACTB, ACTBL2, ACTC1, ANXA1, CADM1, CAST, CDH2, CFDP1, EIF4G2, PFN1, SRRM2 and VCL) as well as MYC transcriptional targets (THBS1 and YWHAZ) (Figure 1D). IPA analyses of the 28 proteins indicated alterations in integrin signaling and actin cytoskeleton signaling among top enriched pathways whereas top predicted molecular functions were related to cellular movement, and cell-to-cell signaling and interaction (Appendix B Table A4), supporting YAP1-associated signatures of cytoskeletal re-arrangement and EMT as prominent features manifest in blood of SCLC cases.

### 3.3. Proteomic Signatures in Plasmas Collected within One Year Prior to Diagnosis of SCLC

We assessed the extent to which proteins identified in plasmas of early stage SCLC cases were similarly manifest in the blood collected within one year prior to diagnosis (Table 2). A total of 1552 plasma proteins were quantified using the same filtering criteria described above (Appendix A).

Among case plasmas collected within one year prior to diagnosis 483 proteins yielded an AUC ≥ 0.60 compared to controls (Figure 2A and Appendix A). Consistent with our findings with samples collected at the time of diagnosis, IPA analysis of proteins with AUC ≥ 0.60 yielded MYC [20] as a central network node with 33 proteins representing transcriptional targets of oncogenic MYC which included several proteins (ANKHD1, ANPEP, CDH12, CDH13, CLASP2, EGFL7, FOXP2, IQGAP3, PAX1, PLEKHA1, THBS1, TPM1, TPX2 and YWHAZ) that are associated with epithelial to mesenchymal transition (EMT) (Appendix B Table A2); YAP1-associated TEAD family members were also found among top predicted upstream regulators (Figure 2B,C) with 61 proteins consisting of transcriptional targets of YAP1 of which several are related to epithelial adherens junction signaling and remodeling (Appendix B Table A3).

Of the proteins with AUC ≥ 0.60 in plasmas of early stage SCLCL cases, 78 were also quantified in the pre-diagnostic specimen set of which 31 were found to be elevated (AUC ≥ 0.60) in plasma of cases diagnosed within one year of blood draw, which included several of YAP1 and MYC-associated proteins (Table 3). Of the proteins that were elevated (AUC ≥ 0.60) in either case plasma sets, there were 52 proteins that were also quantified in SCLC conditioned media (Figure 1D and Figure 2D). Moreover, 9 of the 31 overlapping proteins between the two case sets were quantified in SCLC cell line conditioned medium (Table 3).

### 3.4. Proteomic Findings in Plasmas Collected More Than One Year Prior to DIAGNOSIS

Plasma samples were available that were collected 1-13 years prior to diagnosis. Ten cases in this group were compared to ten controls that were matched for age at the time of blood draw, smoking history and sex. We considered findings from these samples may represent potential risk markers for SCLC given the relatively long span prior to diagnosis. A total of 1242 proteins were quantified using the same criteria as for newly diagnosed cases of which 61 proteins exhibited an odds ratio (OR) of ≥2 with eight proteins exhibiting a 1-sided *p*-value <0.05 (Table 4). Of the proteins with an OR ≥ 2, seven (C9, HCFC2, HYDIN, IGHM, LAMA4, RTKN and TRIM33) were also identified to be elevated (AUC ≥ 0.60) in case plasmas taken at the time of diagnosis or within 0–1 years of diagnosis (Table 4). These proteins are associated with inflammation as in the case of LAMA4 and TRIM33 [38,39].

## 4. Discussion

We report a comprehensive proteomic profiling of case plasmas collected at the time of diagnosis or preceding diagnosis of SCLC with the objective of identifying blood-based markers associated with disease pathogenesis. Our analyses revealed limited occurrence of increased levels of circulating neuroendocrine markers with a notable exception of NSE. However, a notable finding is the occurrence of circulating protein features centered on signatures of oncogenic MYC and YAP1 that were elevated in plasmas of cases at the time-of-diagnosis and at the pre-clinical stage of SCLC. We further report several proteins, with a predominance of inflammatory markers, that were elevated in all case data sets including pre-diagnostic plasmas collected more than one year prior to diagnosis of SCLC and that may be associated with increased risk of disease.

Prior studies have highlighted an essential role of MYC family genes the development of SCLC and in progression towards YAP1-positive states and a mesenchymal phenotype [40,41,42]. YAP1 drives mechanotransduction and remodeling of the cytoskeleton and ECM that occurs during EMT to potentiate cell growth, differentiation and malignant progression [33,34,35,36,37,43]. To this end, among the SCLC-associated biomarkers were several MYC- and YAP1-associated proteins related to EMT and cytoskeletal re-arrangement. EMT-related proteins included YWHAZ and THBS1 that were identified in SCLC cell line conditioned medium and that were elevated in plasmas of newly diagnosed SCLC cases as well as SCLC cases diagnosed within one year of blood draw. Recent evidence has identified a novel function of YWHAZ in activating β-catenin mediated transcription to promote EMT and lung cancer metastasis [44]. Similarly, THBS1 has been shown enhance EMT and promote cancer spreading [45,46]. Among the cytoskeletal-associated proteins that identified in conditioned medium of SCLC cell lines and that were elevated in both early stage and pre-clinical case plasmas were PFN1 and VCL, which have similarly been linked to promoting tumor progression and invasiveness [47,48,49].

We have previously conducted proteomic analyses of plasmas from several mouse models of lung cancer [12]. Among the proteins elevated in plasma of SCLC-bearing mice were ALDOA, CAP1, FBLN2, FGL1, GSR, KRIT1, LTBP4, PDIA3, PKP3, PSMA5, S100A8, SH3BGL3, TPM4 and YWHAE, which were also identified as elevated in case plasmas (AUC ≥ 0.60) in this study, thus providing additional validation of the association between these circulating protein features and SCLC development.

It is noteworthy that several of the protein features found to be elevated in blood taken at the time of diagnosis or preceding diagnosis of SCLC are previously annotated as nuclear proteins. This may be attributed to their release from apoptotic cancer cells, membrane shedding, or occurrence in microparticles such as cancer-cell derived extracellular vesicles [50,51].

We acknowledge that our specimen sets were limited with respect to sample size. Nevertheless, our analyses identified 31 proteins that were elevated in plasmas of early stage SCLC case and that were also elevated in plasmas of SCLC cases diagnosed within one year of blood draw. Although NSE was identified to be elevated in case plasmas, other previously reported SCLC associated markers, such as ProGRP, CgA and POMC, were not detected in our specimen sets [13,14,15,16,17,18,19]. This may be attributed to limited expression at early stage. Alternatively, these proteins may be associated with complexes such as autoantibodies or may occur as heavily modified proteins (e.g., protein glycosylation) [52] that would limit their identification by mass spectrometry.

## 5. Conclusions

In NLST, CT-based screening was not effective for detecting early stage SCLC and no survival improvement was found amongst these individuals [7]. Data from this study revealed several novel circulating protein features that manifest in the blood of early stage SCLC cases and that are associated with oncogenic drivers of SCLC. Candidate biomarkers described herein may have clinical utility for identifying individuals that are at high-risk of developing or harboring SCLC and that would benefit from more intensive follow-up. Further independent validation of candidate proteins is warranted to determine their utility for earlier detection of SCLC.

## Figures and Tables

**Figure 1 cancers-13-03972-f001:**
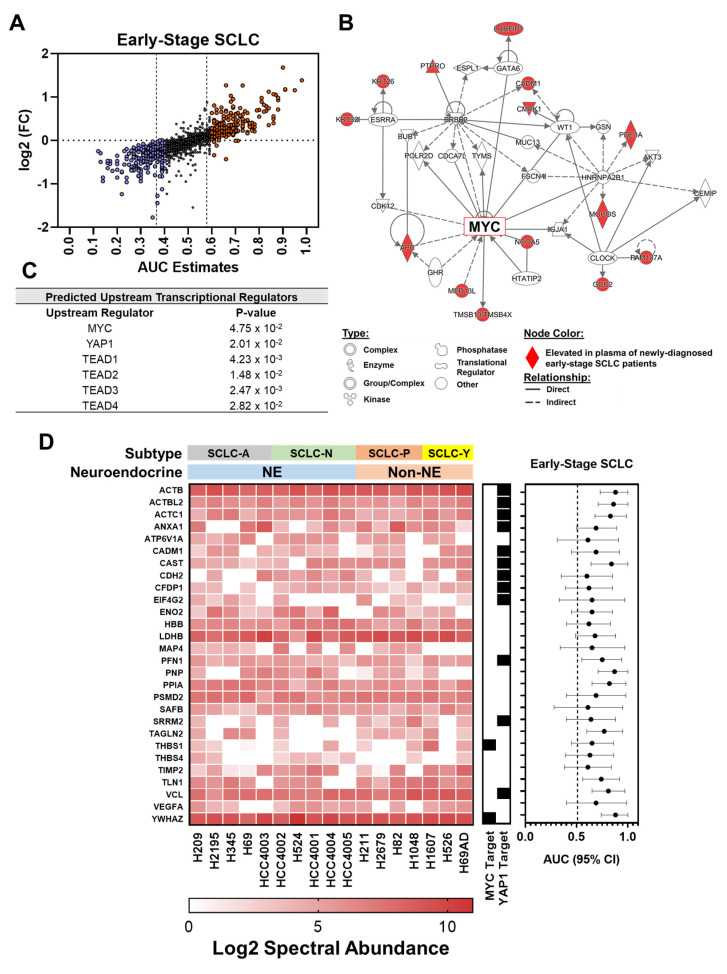
Proteomic profiling of newly diagnosed early stage SCLC patient plasmas reveals signatures associated with oncogenic drivers. (**A**) Distribution plot comparing illustrating the AUC (*x*-axis) and fold-change (*y*-axis) of circulating proteins for distinguishing early stage SCLC cases from matched controls. (**B**,**C**) Ingenuity Pathway Analyses (IPA) on circulating proteins that exhibited an AUC ≥ 0.60 for distinguishing early stage SCLC cases from matched controls reveals MYC (**B**) and YAP1-centric signatures (**C**). (**D**) Heatmap illustrating the 28 proteins that were quantified in conditioned medium of SCLC cell lines (*n* = 17) and that were elevated in plasmas of early stage SCLC cases. The center plot annotates whether the respective protein is a reported downstream transcriptional target (black square) of MYC or YAP1. ChIP Enrichment Analysis (ChEA) transcriptional targets of c-MYC and YAP1 were downloaded from Harmonizome database [29,30]. The scatter plot on the right depicts the AUC (95% CI) of respective proteins for distinguishing early stage SCLC cases from matched controls in the MDACC cohort.

**Figure 2 cancers-13-03972-f002:**
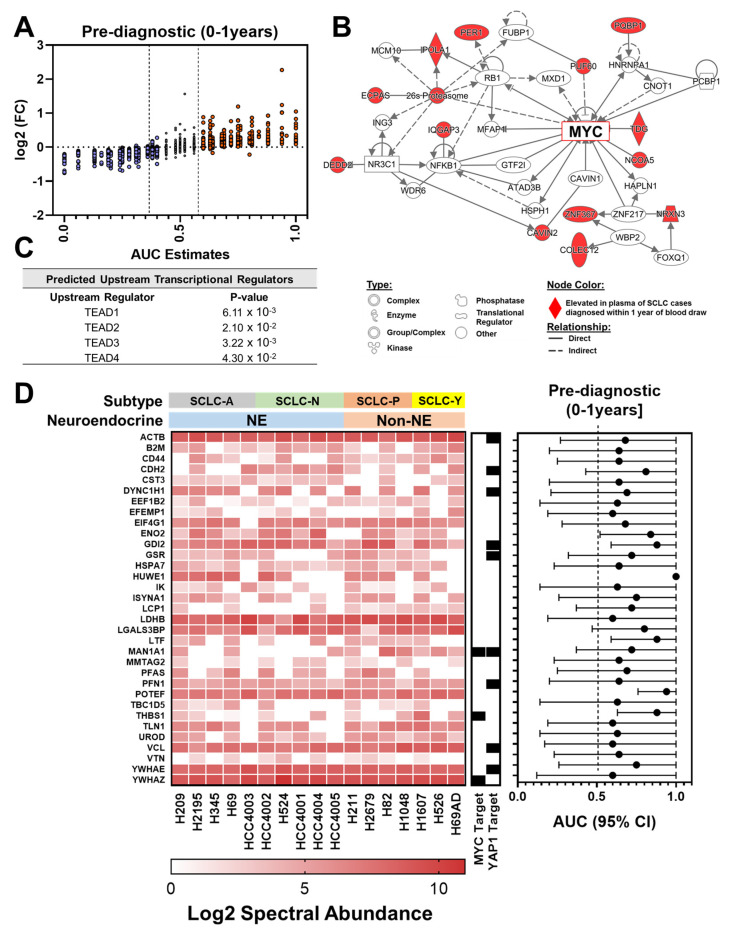
Proteomic signatures of MYC and YAP1 manifest in plasmas collected within one year prior to diagnosis of SCLC. (**A**) Distribution plot comparing illustrating the AUC (*x*-axis) and fold-change (*y*-axis) of circulating proteins for distinguishing SCLC cases diagnosed within one year of blood draw from matched controls. (**B**,**C**) Ingenuity Pathway Analyses (IPA) on circulating proteins that exhibited an AUC ≥ 0.60 for distinguishing SCLC cases diagnosed within one year of blood draw from matched controls reveals MYC (**B**) and YAP1-centric signatures (**C**). (**D**) Heatmap illustrating the 33 proteins that were quantified in conditioned medium of SCLC cell lines (*n* = 17) and that were elevated in plasmas of SCLC cases diagnosed within one year of blood draw. The center plot annotates whether the respective protein is a reported downstream transcriptional target (black square) of MYC or YAP1. ChIP Enrichment Analysis (ChEA) transcriptional targets of c-MYC and YAP1 were downloaded from Harmonizome database [29,30]. The scatter plot on the right depicts the AUC (95% CI) of respective proteins for distinguishing SCLC cases from matched controls in the pre-diagnostic (0–1 year] dataset.

**Table 1 cancers-13-03972-t001:** Patient and tumor characteristics for MDACC Cohort.

Patient and Tumor Characteristics	Cases	Controls	*p* ^†^
*N*	15	15	
Age, mean ± stdev	67 ± 10	64 ± 5	0.330
Sex, *N* (%)			
Male	8 (53.3%)	8 (53.3%)	
Female	7 (46.7%)	7 (46.7%)	
Stage, *N* (%)			
I	6 (40%)	-	
II	9 (60%)	-	
Smoking PYs, mean ± stdev	63 ± 27	51 ± 18	0.200

^†^ 2-sided student *t*-test.

**Table 2 cancers-13-03972-t002:** Patient and tumor characteristics for the entire pre-diagnostic SCLC Cohort.

Patient and Tumor Characteristics	Cases	Controls
*N*	15	15
Age, mean ± stdev	62.6 ± 8.7	62.5 ± 8.9
Years from Dx, median (min/max)	2.4 (0.7, 12.3)	-
Sex, *N* (%)		
Female	8 (53.3%)	8 (53.3%)
Male	7 (46.7%)	7 (46.7%)
Smoking Status, *N* (%)		
Former	2 (13.3%)	2 (13.3%)
Current	12 (80.0%)	12 (80.0%)
Never	1 (6.7%)	1 (6.7%)

**Table 3 cancers-13-03972-t003:** Overlap of proteins elevated (AUC ≥ 0.60) in case plasmas at the time-of-diagnosis and preceding diagnosis of SCLC.

Protein	At-Dx	Pre-Dx	Quantified in SCLC CM ^‡^	MYC Downstream Target ^║^	YAP1 Downstream Target ^║^
	Early Stage SCLC ^†^	(0–1 Year] ^†^			
ACTB	0.88 (0.73–1.00)	0.68 (0.27–1.00)	Yes	-	Yes
C9	0.60 (0.38–0.81)	0.60 (0.19–1.00)	-	-	-
CA1	0.64 (0.43–0.85)	0.60 (0.19–1.00)	-	-	-
CDH2	0.60 (0.35–0.85)	0.81 (0.43–1.00)	Yes	-	Yes
COL6A6	0.61 (0.3–0.92)	0.64 (0.23–1.00)	-	-	-
CPT2	0.78 (0.52–1.00)	0.64 (0.20–1.00)	-	-	Yes
CRP	0.68 (0.49–0.88)	0.64 (0.25–1.00)	-	-	-
D2HGDH	0.66 (0.46–0.87)	0.63 (0.14–1.00)	-	-	-
ENO2	0.65 (0.45–0.85)	0.84 (0.52–1.00)	Yes	-	-
KIF27	0.64 (0.34–0.94)	0.75 (0.26–1.00)	-	-	-
KLHDC10	0.62 (0.30–0.94)	0.63 (0.14–1.00)	-	-	Yes
LDHB	0.68 (0.49–0.88)	0.60 (0.19–1.00)	Yes	-	-
LONRF1	0.86 (0.66–1.00)	0.94 (0.76–1.00)	-	-	-
MLPH	0.75 (0.49–1.00)	0.69 (0.21–1.00)	-	-	-
MYD88	0.70 (0.48–0.92)	0.76 (0.41–1.00)	-	-	-
NCOA5	0.63 (0.37–0.88)	0.60 (0.19–1.00)	-	-	-
OLFML2A	0.66 (0.36–0.95)	1.00 (1.00–1.00)	-	-	-
OPLAH	0.60 (0.36–0.85)	0.60 (0.21–0.99)	-	-	-
PER1	0.65 (0.31–1.00)	0.64 (0.25–1.00)	-	Yes	-
PFN1	0.75 (0.55–0.94)	0.64 (0.20–1.00)	Yes	-	Yes
PPBP	0.62 (0.41–0.83)	0.76 (0.41–1.00)	-	-	-
RAB17	0.61 (0.28–0.94)	0.64 (0.25–1.00)	-	-	-
S100A12	0.73 (0.54–0.92)	0.94 (0.76–1.00)	-	-	-
S100A8	0.69 (0.49–0.89)	0.72 (0.32–1.00)	-	-	-
THBS1	0.65 (0.45–0.86)	0.88 (0.63–1.00)	Yes	Yes	-
TLN1	0.74 (0.56–0.92)	0.60 (0.19–1.00)	Yes	-	-
TRRAP	0.66 (0.36–0.95)	0.63 (0.14–1.00)	-	-	-
UNC80	0.65 (0.42–0.89)	0.81 (0.43–1.00)	-	-	-
USP4	0.60 (0.38–0.81)	0.94 (0.76–1.00)	-	-	-
VCL	0.81 (0.65–0.97)	0.60 (0.17–1.00)	Yes	-	Yes
YWHAZ	0.88 (0.74–1.00)	0.60 (0.12–1.00)	Yes	Yes	-

^†^ AUC (95% confidence interval) for distinguishing cases from matched controls. ^‡^ Proteins quantified in conditioned medium of SCLC cell lines are provided in Appendix A. ^║^ ChIP Enrichment Analysis (ChEA) transcriptional targets of c-MYC and YAP1 were derived from Harmonizome database.

**Table 4 cancers-13-03972-t004:** The 61 circulating proteins that exhibited an Odds Ratio >2 for SCLC lung cancer.

Protein	Cases (Mean ± SD)	Controls (Mean ± SD)	Odds Ratio ^†^	1-Sided *p*-Value	Elevated (AUC ≥ 0.60) in Case Plasmas ^‡^
ACSL1	1.45 ± 0.28	1.3 ± 0.15	2.24	0.1000	
AJM1	1.57 ± 0.65	1.27 ± 0.34	2.01	0.0880	
AKAP9	2.2 ± 0.63	1.92 ± 0.35	2.02	0.1350	
ALPI	1.26 ± 0.09	1.17 ± 0.14	2.60	0.0630	
APOD	1.52 ± 1.39	1.05 ± 0.22	2.59	0.1110	
C4BPA	1.28 ± 0.17	1.14 ± 0.24	2.21	0.0580	
C9	1.18 ± 0.14	1.05 ± 0.22	2.24	0.0550	✓
CARD6	1.38 ± 1.29	0.93 ± 0.15	2.36	0.1430	
CCDC115	1.12 ± 0.14	1.01 ± 0.06	9.43	0.0230	
CDHR1	1.93 ± 2.24	1.05 ± 0.15	25.60	0.0960	
CDKL1	1.09 ± 0.1	1.03 ± 0.07	2.18	0.0950	
CTNND2	1.16 ± 0.29	0.98 ± 0.17	2.35	0.0800	
CTPS1	1.57 ± 0.83	1.25 ± 0.24	2.11	0.1520	
DDX31	0.88 ± 0.21	0.71 ± 0.08	10.87	0.0110	
ERCC4	1.76 ± 1.11	1.22 ± 0.39	2.97	0.1000	
FBXL8	1.09 ± 0.17	0.98 ± 0.11	2.37	0.0430	
FGA	2.07 ± 3.5	0.93 ± 0.21	4.74	0.1050	
FGB	1.83 ± 2.6	1.02 ± 0.2	2.32	0.1290	
FGG	2.25 ± 3.88	0.95 ± 0.2	13.01	0.0890	
FUCA1	1.17 ± 0.23	1.01 ± 0.14	2.85	0.0520	
HCFC2	1.35 ± 0.2	1.25 ± 0.06	2.72	0.0820	✓
HEXD	1.83 ± 2.23	0.95 ± 0.22	4.98	0.1050	
HYDIN	1.11 ± 0.26	0.96 ± 0.16	2.43	0.0850	✓
IGHM	1.94 ± 0.99	1.39 ± 0.41	2.49	0.0460	✓
JMJD1C	0.73 ± 0.13	0.65 ± 0.1	2.27	0.1040	
KIFC2	2.66 ± 1.96	1.74 ± 0.45	5.02	0.0730	
LAMA4	1.14 ± 0.29	0.97 ± 0.2	2.22	0.0910	✓
LENG9	1.41 ± 0.16	1.28 ± 0.19	2.25	0.0860	
MAP3K7	1.09 ± 0.18	1.01 ± 0.04	3.84	0.0910	
MAPKAPK5-AS1	1.11 ± 0.5	0.87 ± 0.16	2.87	0.0510	
MBD5	1 ± 0.09	0.93 ± 0.08	2.84	0.0480	
MOV10L1	1.24 ± 0.16	1.13 ± 0.21	2.03	0.1380	
MX1	1.23 ± 0.09	1.15 ± 0.12	2.34	0.0870	
NACA	1.46 ± 0.18	1.35 ± 0.2	2.00	0.1230	
NLN	1.21 ± 0.11	1.13 ± 0.12	2.27	0.0850	
NUCB2	1.24 ± 0.12	1.14 ± 0.18	2.16	0.0960	
OTUD6A	0.9 ± 0.11	0.83 ± 0.09	2.31	0.0840	
PARD3	1.4 ± 0.62	1.14 ± 0.22	2.26	0.1390	
PCNT	1.11 ± 0.28	0.9 ± 0.27	2.56	0.0640	
PDE4DIP	1.74 ± 1.12	1.28 ± 0.35	2.28	0.1230	
PHC1	0.84 ± 0.31	0.7 ± 0.04	3.05	0.1110	
PHRF1	1.03 ± 0.48	0.84 ± 0.17	2.11	0.1320	
PIEZO1	1.34 ± 0.24	1.19 ± 0.18	2.27	0.1090	
PLEC	1.29 ± 0.24	1.09 ± 0.25	2.85	0.0520	
PML	1.15 ± 0.12	0.94 ± 0.29	3.36	0.0460	
POLR1B	3.17 ± 0.7	2.74 ± 0.62	2.23	0.1020	
RBFA	1.26 ± 0.37	1.04 ± 0.29	2.06	0.1060	
RCCD1	1.15 ± 0.1	1.05 ± 0.09	6.71	0.0220	
RNASE4	1.29 ± 0.67	1.05 ± 0.16	2.58	0.0970	
RTKN	1.3 ± 0.62	1.07 ± 0.09	2.41	0.1440	✓
SLC4A10	1.51 ± 0.33	1.25 ± 0.2	3.75	0.0320	
SLC6A15	1.22 ± 0.41	1.05 ± 0.06	2.75	0.1200	
SPANXA2-OT1	1.78 ± 1.47	1.35 ± 0.15	2.01	0.1480	
THRAP3	1.17 ± 0.19	1.04 ± 0.18	2.20	0.0920	
TNKS1BP1	0.96 ± 0.07	0.92 ± 0.05	2.12	0.1120	
TNNI3K	2.36 ± 1.6	1.71 ± 0.24	8.87	0.0500	
TRIM33	2.17 ± 4.2	0.73 ± 0.25	38.99	0.0720	✓
TTC6	1.12 ± 0.86	0.79 ± 0.14	2.65	0.1280	
VPS13C	1.39 ± 0.44	1.21 ± 0.13	2.41	0.1330	
WDR44	1.48 ± 1.59	0.92 ± 0.12	2.42	0.1440	
WDR46	1.24 ± 0.41	0.95 ± 0.32	2.83	0.0570	

^†^ Per unit increase. ^‡^ Designates that the respective protein was also found to be elevated (AUC ≥ 0.60) in plasmas of newly diagnosed early stage SCLC cases or SCLC cases diagnosed within 1 year of blood draw relative to matched controls.

## Data Availability

Relevant data supporting the findings of this study are available within the Article, the Appendix B and Appendix A. Information or are available from the authors upon reasonable request.

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
