# Peer review of "Plasma Based Protein Signatures Associated with Small Cell Lung Cancer"

_cancers, 2021, doi:10.3390/cancers13163972_

Round 1
Reviewer 1 Report
The manuscript entitled:" Plasma Based Protein Signatures Associated with Small Cell Lung Cancer " focused on the evaluation of protein signature in the early stage of SCLC patients is well written and requires some minor considerations to be accepted for the publication
- In the introduction section, the authors should better clarify the issues in the early detection of lung cancer patients by elucidating the limitations of currently available biomarkers for this clinical setting.
- In the discussion section, i would reccomend to clarify how this approach may be used in the clinical administration of SCLC patients.
Author Response
The manuscript entitled:" Plasma Based Protein Signatures Associated with Small Cell Lung Cancer " focused on the evaluation of protein signature in the early stage of SCLC patients is well written and requires some minor considerations to be accepted for the publication
-We thank the Reviewer for their constructive and overall positive remarks.
In the introduction section, the authors should better clarify the issues in the early detection of lung cancer patients by elucidating the limitations of currently available biomarkers for this clinical setting.
Response: We have now provided additional content regarding issues related to early detection of lung cancer in the introduction section of the revised manuscript.
In the discussion section, i would recommend to clarify how this approach may be used in the clinical administration of SCLC patients.
Response: In NLST, CT-based screening was not effective for detecting early-stage SCLC and no survival improvement was found amongst these individuals. Candidate biomarkers described in our report may have clinical utility for identifying individuals that are at high-risk of developing or harboring SCLC and that would benefit from more intensive follow-up. Further independent validation of candidate proteins is warranted to determine their utility for earlier detection of SCLC.
We have now commented upon this in the conclusion section of the revised manuscript.
Reviewer 2 Report
The authors of the article "Plasma Based Protein Signatures Associated with Small Cell 2 Lung Cancer" did an excellent job of discovering different circulating markers that can be utilized to detect SCLC at an early stage. However, there are a few issues that need to be addressed before the article can be published.
The introduction appears to be very brief and insufficient.
Some proteins (such as ZNFs in the supplimentory) were discovered in the proteomics research are neuclear and not supposed to be in plasma. In terms of technical contamination, this makes the findings a little perplexing. Could any of these proteins be a result of cell lysis during the plasma collection process? What are the authors' thoughts on the subject?
Author Response
The authors of the article "Plasma Based Protein Signatures Associated with Small Cell Lung Cancer" did an excellent job of discovering different circulating markers that can be utilized to detect SCLC at an early stage. However, there are a few issues that need to be addressed before the article can be published.
-We thank the Reviewer for their support of our study and findings.
The introduction appears to be very brief and insufficient.
Response: We have now expanded our introduction to better emphasis the importance of our study in relation to the need for biomarkers for early detection of small cell lung cancer.
Some proteins (such as ZNFs in the supplementary) were discovered in the proteomics research are neuclear and not supposed to be in plasma. In terms of technical contamination, this makes the findings a little perplexing. Could any of these proteins be a result of cell lysis during the plasma collection process? What are the authors' thoughts on the subject?
Response: The Reviewer is accurate that several of the proteins elevated in blood taken at the time of diagnosis or preceding diagnosis of SCLC are annotated as nuclear proteins. This may be attributed to release from apoptotic cancer cells, membrane shedding, or occurrence in microparticles such as cancer-cell derived extracellular vesicles. [1,2] We have now commented upon this in the discussion section of the revised manuscript.
References
- Vykoukal, J.; Sun, N.; Aguilar-Bonavides, C.; Katayama, H.; Tanaka, I.; Fahrmann, J.F.; Capello, M.; Fujimoto, J.; Aguilar, M.; Wistuba, II; et al. Plasma-derived extracellular vesicle proteins as a source of biomarkers for lung adenocarcinoma. Oncotarget 2017, 8, 95466-95480, doi:10.18632/oncotarget.20748.
- Fahrmann, J.F.; Mao, X.; Irajizad, E.; Katayama, H.; Capello, M.; Tanaka, I.; Kato, T.; Wistuba, II; Maitra, A.; Ostrin, E.J.; et al. Plasma-Derived Extracellular Vesicles Convey Protein Signatures that Reflect Pathophysiology in Lung and Pancreatic Adenocarcinomas. Cancers (Basel) 2020, 12, doi:10.3390/cancers12051147.